# The Impact of Voluntary Policies on Parents’ Ability to Select Healthy Foods in Supermarkets: A Qualitative Study of Australian Parental Views

**DOI:** 10.3390/ijerph16183377

**Published:** 2019-09-12

**Authors:** Claire Elizabeth Pulker, Denise Chew Ching Li, Jane Anne Scott, Christina Mary Pollard

**Affiliations:** 1School of Public Health, Curtin University, Kent Street, GPO Box U1987, Perth 6845, Western Australia, Australia; denisechew13@gmail.com (D.C.C.L.); jane.scott@curtin.edu.au (J.A.S.); c.pollard@curtin.edu.au (C.M.P.); 2East Metropolitan Health Service, Kirkman House, 20 Murray Street, East Perth 6004, Western Australia, Australia

**Keywords:** food choice, food decision making, food label, marketing, supermarket, food policy, children

## Abstract

Food packaging is used for marketing purposes, providing consumers with information about product attributes at the point-of-sale and thus influencing food choice. The Australian government focuses on voluntary policies to address inappropriate food marketing, including the Health Star Rating nutrition label. This research explored the way marketing via packaging information influences Australian parents’ ability to select healthy foods for their children, and who parents believe should be responsible for helping them. Five 90-min focus groups were conducted by an experienced facilitator in Perth, Western Australia. Four fathers and 33 mothers of children aged 2–8 years participated. Group discussions were audio-recorded and transcribed verbatim and inductive thematic content analysis conducted using NVivo11. Seven themes were derived: (1) pressure of meeting multiple demands; (2) desire to speed up shopping; (3) feeding them well versus keeping them happy; (4) lack of certainty in packaging information; (5) government is trusted and should take charge; (6) food manufacturers’ health messages are not trusted; (7) supermarkets should assist parents to select healthy foods. Food packaging information appears to be contributing to parents’ uncertainty regarding healthy food choices. Supermarkets could respond to parents’ trust in them by implementing structural policies, providing shopping environments that support and encourage healthy food choices.

## 1. Introduction

Powerful food companies have been identified as drivers of obesity and diet-related non-communicable diseases [1]. Poor diet is one of the most important risk factors for early deaths globally [2]. In Australia, supermarkets act as primary gatekeepers of the food system [3], and there is a high level of foreign ownership of food brands by large global manufacturers [4]. Food manufacturers’ products and marketing activities contribute to shaping food environments [5], which have a substantial impact on food selection and diet [6]. The Australian Guide to Healthy Eating specifies five food groups that are essential for good health (i.e., healthy), and discretionary foods that should be limited (i.e., unhealthy) [7]. Although supermarkets sell healthy food, in Australia, less than half of commonly available supermarket packaged foods were classified as healthy using the Food Standards Australia New Zealand nutrient profiling scoring criterion [8]. Unhealthy snack foods such as crisps and confectionery are displayed at prominent supermarket locations, such as the ends-of-aisles and checkouts [9].

Excessive marketing of unhealthy foods presents a major threat to public health, particularly for children [10]. Packaging, a key marketing method, is the primary means of communicating information to consumers about product attributes at the point-of-sale [11,12]. A large proportion of supermarket purchases are made on impulse, and packaging plays a crucial role in purchasing decisions [13]. Packaged foods designed to appeal to children are widely available and displayed in prominent supermarket locations [14]. Sixteen marketing techniques have been used on packaging to appeal to children including cartoons and celebrities, and most products marketed to children via packaging are unhealthy [13]. Marketing techniques include use of nutrition-related statements and claims on unhealthy foods, which parents interpret as being more nutritious than foods without claims [15]. Excessive marketing of these nutrient-poor discretionary foods, or ‘fun foods’ [16], to children encourages overconsumption [17], which is associated with increasing prevalence of childhood obesity [18]. Few Australian children consume diets consistent with Australian Dietary Guideline recommendations [19]. Australian parents are concerned about food marketing to children [20] and believe it influences their children’s food preferences [21]. Policies that address the information provided on food packaging are needed to assist parents to select healthy foods [22].

Government responses to food marketing targeting children have been driven by the dominant neoliberal political agenda in many countries, whereby policy is minimised to promote global trade [23]. Neoliberal governance supports voluntary policies and industry self-regulation, as it requires the least government intervention [24]. The Australian government’s focus is on voluntary policies to assist consumers to make healthier food choices, including the Health Star Rating front-of-package interpretive labelling system (HSR), which was launched in 2014 (Figure 1) [25], and the Healthy Food Partnership public-private-partnership, which aims to improve the nutritional health of all Australians [26]. Public health responses often focus on providing information and education to assist individuals to select healthy foods [27]; however, policy and regulations can alter population environments [28] and tend to be more effective, rapid and equitable [27]. There is a lack of research examining consumer use of front-of-pack nutrition labelling in real world settings [29].

Food companies, including manufacturers and retailers, have the collective power to assist consumers to select healthy foods [31]. Voluntary policies and participation in industry self-regulation are framed by food companies as socially responsible initiatives designed to ensure consumer welfare. Australia’s Responsible Children’s Marketing Initiative has not addressed marketing at point-of-sale including packaging to date [32], which has been identified as a limitation of such approaches [33]. Voluntary policies have also been criticised as a mechanism to differentiate products and appeal to consumers [34], pass responsibility from food companies to consumers [35] and prevent regulation [36]. Although consumers may not be aware of the policies, they select food in supermarket environments that are impacted by food companies’ efforts to act responsibly.

A review of Australian food companies’ policies and commitments related to obesity prevention found many did not have policies on marketing food to children and, where present, restrictions were not strong enough to be effective [37,38]. Voluntary policies have been used to build brand reputation, appeal to parents and children through community activities and align with respected organisations [39], which was valued by parents and children [40].

High-market-share Australian food companies have made statements about the importance of health and nutrition (see Table 1 for examples). However, the ability of such statements and voluntary policies to assist consumers to select healthy foods consistent with the recommendations of national dietary guidelines has not been investigated. The Australian government-led voluntary HSR was designed to guide selection of healthier packaged foods, by providing a single, consistent, accurate, front-of-pack nutrition label [41]. To date, no study has specifically explored how information-based voluntary policies, such as the HSR, influence parents’ food selection. It is not known who parents believe should be responsible for providing the information they need to select healthy foods, or the relative value they place on information provided by food manufacturers, supermarkets or government. Parents’ use and understanding of HSR and other nutrition messages in real world settings can inform policy recommendations for minimizing harm from inappropriate food marketing to children.

This study aimed to describe the impact of voluntary policies on parents’ ability to select healthy foods in supermarkets by exploring the way food packaging information influences their choices, and who parents believe should be responsible for helping them. Due to the business nature of voluntary policies, the specific research questions did not refer to the concept directly. The research questions are: (1) Are parents able to navigate marketing techniques used on packaging to select healthy foods for their children? (2) Who do parents believe should be responsible for giving them the information they need to make healthy food choices for their children? (3) What role do parents believe food companies should take in helping them select healthy foods for their children?

## 2. Materials and Methods

The exploratory nature of this research required a qualitative approach to encourage open-ended in-depth inquiry of the topic [47]. The research was informed by social constructivism, which relates to how knowledge is constructed and understood by social groups rather than individuals [48]. The paradigm of social constructivism was applied to this research to understand the lived experience of parents of young children from their own perspective. Therefore, focus groups were chosen to allow participants to talk to each other as well as the facilitator, which is a useful way of exploring knowledge and experience [49]. Group interaction can help participants clarify and compare views or issues and highlight cultural norms, and are particularly useful when examining attitudes [49]. However, as focus groups identify group norms and knowledge [50], there is a risk that participants are unlikely to challenge the dominant group perspective [49]. The consolidated criteria for reporting qualitative research (COREQ) were used to describe the methods for this study [51] (Appendix A). The Curtin University Human Research Ethics Committee approved the research study (RDHS-186-15).

### 2.1. Participants and Recruitment

Parents of young children (2–8 years) were recruited from a market research panel of adults who had expressed interest in contributing to research. Parents with children of this age were the focus of the study given their influence on young children’s food choices and preferences. Purposive sampling was used to recruit participants from both high and low socio-economic status (SES) areas [52] to ensure the views of parents from different SES backgrounds were represented. Participants were assigned to focus groups based on their SES classification and age of youngest child to encourage free expression of any challenges faced based on these characteristics. Focus groups generate research data when participants communicate with each other about the research topic [49]. Commonality of experience is an important component of building trust among focus group participants [53]; therefore, running separate groups on the basis of SES was considered consistent with best practice in focus group methodology.

Potential participants were contacted by a market research company by telephone if they met the screening criteria (main household food shopper; had children aged 2–8 years; no immediate family working for a food manufacturer or in market research, advertising or nutrition) and invited to attend a focus group. Participants were given AUD$80 as compensation for their time and travel costs.

Participants (*N* = 37) comprised of four fathers and 33 mothers aged 25–48 years, most of Caucasian descent. Most participants from the first, second and fifth groups lived in high SES areas, and those in the third and fourth groups lived in lower SES areas (Table 2).

### 2.2. Design and Procedure

Five 90-min focus groups were conducted in 2015. On arrival, researchers gave participants an information sheet stating the purpose, risk and benefits of the study, informed them that the focus group would be recorded and they would be de-identified during transcription to ensure all data were anonymous. Written consent for the recording to take place was obtained. Demographic data were collected using a written survey prior to the groups commencing. Focus groups were conducted at a market research company, in a conference room with a large table and chairs and light refreshments were provided. An experienced qualitative researcher employed by a market research company facilitated the focus groups. The independent researcher had no prior experience related to the topic under investigation. Whilst detailed content knowledge is preferred for some focus group discussions [53], facilitation of the focus groups by a professional moderator minimised risk of bias with predetermined views in this study. Two researchers were present in a room adjacent to the conference room and observed the focus group discussions via a two-way mirror.

### 2.3. Selection of Visual Stimuli

Use of visual stimuli to encourage discussion between focus group participants has been shown to be effective [54]. Focus group activities where participants are asked to sort, rank or list are referred to as activity-orientated questions, and can be a useful way of promoting discussion [55]. Twenty-five packaged foods were introduced to stimulate discussion (Appendix A). Products were selected from high-market share food manufacturers to increase the likelihood of familiarity. They were chosen to show a variety of marketing techniques commonly used by manufacturers to appeal to children, including cartoon characters, unusual shapes and bright colours [13]. The packaging also featured a wide range of health statements and claims, including nutrition claims, health claims and the HSR nutrition label with a range of scores. Each of the selected high-market share food manufacturers made statements about the importance of nutrition and health (Table 1), which have the potential to limit use of inappropriate health statements and claims and marketing techniques on unhealthy foods. Fresh apples and bananas acted as benchmarks for healthy foods.

### 2.4. Development of Focus Group Discussion Guide

A semi-structured guide directed focus group discussions, whilst allowing for diversions reflective of participants’ statements (Table 3). The concept of voluntary policies was considered too abstract to address directly in the focus groups. The guide was designed to prompt discussion about the way food packaging might influence participants when shopping, so that the impact of policies which aim to support consumers to select healthy foods and limit inappropriate marketing of unhealthy foods could be explored (Table 1). The guide also prompted discussion about the relative value participants placed on information provided by food manufacturers, supermarkets and government. The guide had seven questions, starting with an icebreaker to encourage free expression of the difficulties of feeding young children that prompted participants to introduce themselves and their families and rate each family member’s food fussiness. The first focus group was used as a pilot test of the discussion guide; however, the researchers were satisfied that no changes were required.

To address the first research question, participants were asked to work together as a group to sort the 25 products into groups of similar products, to understand how they categorised foods, used packaging information to make decisions and whether there was general agreement on criteria (if used). It was anticipated that the product sorting exercise would identify decision-informing criteria related to price, child-orientation, appetite appeal, acceptability and/or health. If health was not chosen, the facilitator later prompted the group to sort the products again according to perceived healthiness.

The facilitator then asked a number of questions to address the second research question. Participants were prompted to think about the information they use to decide whether products were healthy or not, who they think is responsible for providing information on packaged foods and whether government, food manufacturers or supermarkets should be responsible for helping parents choose healthier foods.

To address the third research question, participants were asked what food manufacturers and supermarkets could do to assist them to select healthy food. During the product sorting exercise, they were prompted to discuss their views on the information provided on food packaging and whether or not it was helpful in making healthy choices.

### 2.5. Data Analysis

Data were analysed concurrently with focus group data collection to determine when saturation was reached [51]. Audiotapes were transcribed verbatim by a professional transcription service, and data were entered into NVivo11(QSR International Pty Ltd., Melbourne, VIC, Australia) and reviewed line by line for concepts. Inductive content analysis of the transcripts was conducted by two researchers concurrently, which allowed patterns and themes to surface. The process included initial familiarization with the transcripts, followed by open coding of text segments that addressed the research questions. Coded text segments that expressed similar parental views were then grouped together as themes. Review of the coded text segments continued until overlap among the themes was reduced, and the main themes identified [56]. Two researchers conducted the content analysis independently, to identify whether there was consistency in the themes they identified, which is an important check of interrater reliability and reduces bias [56]. When general consistency in the themes was achieved between the two researchers, the list of themes relevant to the research questions were reviewed by all of the research team, and the most important themes were discussed and agreed.

### 2.6. Analytical Framework

The implications for voluntary policies are discussed applying a political (i.e., large companies accept responsibility for their impact on society via corporate citizenship) and ethical (i.e., companies accept social responsibilities as an ethical obligation) lens to analysis of findings [57].

## 3. Results

The seven most important themes inductively derived from analysis of the transcripts were grouped into two broad categories. Firstly, themes relating to parents’ ability to navigate the marketing information present on packaging to select healthy foods: (1) pressure of meeting multiple demands; (2) desire to speed up shopping; (3) feeding them well versus keeping them happy; (4) lack of certainty in packaging information. Secondly, themes relating to who participants thought should be responsible for assisting parents to select healthy foods: (5) government is trusted and should take charge; (6) food manufacturers’ health messages are not trusted; (7) supermarkets should assist parents to select healthy foods. Although the majority of parents were from medium to high SES neighbourhoods, the themes identified from the two focus groups consisting of predominantly low SES parents did not differ from the themes identified from focus groups of parents of higher SES. Each theme is explored below, using illustrative quotes from participants to reflect the shared perspectives of each group.

### 3.1. Pressure of Meeting Multiple Demands

Participants described how they attempted to meet all of the different food tastes and preferences of their family, and avoid wasting food, while trying to ensure their children ate a varied diet. One participant in group 3 described the pressure she felt many mothers experienced:
“We have so much pressure I think these days to be a ‘good mum’ and even when I’m standing at the checkout I worry what are people going to think about what’s in my trolley”(group 3 participant)

Many of the participants described preparing multiple versions of a meal, or preparing food in advance and freezing it. Meeting the needs of children that were fussy eaters, by preparing more than one main meal for the family was discussed by participants in groups 1, 2 and 4. Participants in groups 1 and 2 described how they did not want to buy food for their children that they would reject, meaning it would be wasted:
“Half of it’s if they’re going to eat it or not as well, because I don’t know about everyone else but wasting food is bad you know, especially when it’s as expensive as it is”(group 2 participant)
“I may as well just throw my money in the bin”(group 1 participant)

One participant in group 4 referred to parents of children who would eat anything as “lucky”, and then described how they prepared three family meals each day: one for the parents, one for their daughter with coeliac disease and one for their fussy young son. Other participants agreed, with one describing how she met the multiple demands of her family:
“We’ll do spag bol (sic), it’ll have vegetables that are big enough that you can take them out for one kid and then noodles, probably have noodles because my daughter only wants noodles, and then I’ll have no pasta and I’ll just have it with broccoli”(group 4 participant)

All focus groups discussed how participants selected specific foods for school lunchboxes, which required careful consideration. Participants described the rules set by schools for permitted lunchbox foods, as well as practical considerations such as food remaining safe until lunchtime, and fostering independent eating for younger children by ensuring they were able to open food packages themselves. Some of the schools required children to bring one piece of fruit each day (discussed in group 1), some schools did not allow children to eat products containing nuts (discussed in group 5) and some schools banned specific foods such as squeezable yoghurt pouches because children made a mess with them (discussed in group 2). In addition, schools rewarded healthy lunchboxes by giving children a ‘I am a healthy eater’ sticker (group 3 participant), or gave children reward tokens when they brought lunchboxes containing no commercial packaging in an effort to reduce waste (group 4 participant).

Participants in groups 3 and 5 described how the foods their children took to school differed to foods eaten at home, which was summed up by one participant:
“They are not going to take a meal to have on a plate, it’s stuff that’s got to be in their lunchbox and it’s got to survive”(group 3 participant)

Overall, eating healthily was considered laborious and time-consuming, as summarised by one participant:
“Depends if you’ve got the time actually, and the inclination to actually compare what’s in the different brands”(group 4 participant)

Participants in group 5 discussed the challenge of feeding fussy children who would suddenly refuse to eat previously accepted foods, with one describing how she has “a bit of an arsenal” of foods as back up to ensure her son ate healthily. Participants in all groups discussed how reading nutrition labels while shopping with children, and preparing and cooking healthy meals, was difficult to incorporate into busy schedules.
“You have to be organised I think, if you want to eat healthy you’ve got to be prepared and organised and know exactly what you’re going to have”(group 5 participant)

### 3.2. Desire to Speed up Shopping

Parents of younger children emphasised the desire to get in and out of the supermarket as fast as possible, leaving little time to scrutinise food labels. To speed up the shopping process, participants from all focus groups discussed how they used their own criteria to determine whether foods were appropriate, including: avoiding foods with colours, flavours, preservatives; avoiding sugary foods; avoiding specific areas of the supermarket such as the confectionery aisle. Sugar was a common ingredient that participants tried to avoid, which was discussed by all focus groups. The discussion group 1 participants had about their criteria for selecting healthy foods illustrates the types of information used to select suitable products:
“I try to get things that say low sugar or no preservatives on the packaging”“Or wholegrain”“Yes, no artificial colours, no preservatives all that stuff”(group 1 participants)

Participants in group 3 discussed some of their shortcuts to selecting healthy foods:
“For me it’s the ingredients list, if it’s got too many things on there and I don’t know what they are, I don’t buy it”“Yes, it’s got to be things that I know what they are”“If it doesn’t say ‘light’ on there, I’ll buy it”“If it’s light dairy or low fat dairy it means there’s added sugar, so dairy needs to be full fat”(group 3 participants)

Participants from all of the focus groups assumed that products that appealed to their children, with colourful packaging or licensed characters, were unhealthy choices.
“When you have pictures of princesses or you know they try to attract kids, it’s not healthy for the kids”(group 4 participant)
“I think the brighter they are, the more sugar is in it”(group 2 participant)
“Colours do make a difference because if I saw something like that Rice Bubble thing multi-colours, rainbows, I would think it’s full of sugar”(group 1 participant)

In addition, group 1 participants discussed how they did not want to encourage their children to find the packaging appealing, and thus avoided such foods for these additional reasons.

The challenges of avoiding foods designed to appeal to children were identified in group 2, who described how these foods were often the ones their young children would be most likely to eat:
“Anything that has a princess picture my daughter will eat”(group 2 participant)

However, one participant said she did not look for specific cues on packaging to make healthy choices for her family:
“I just trust my instinct, because we’ve survived this long”(group 2 participant)

Despite discussion about HSR in all of the groups, and its presence on half of the packaging stimuli, participants did not discuss the HSR label as a tool to assist them to speed up shopping. In some groups, it appeared that the lack of use of HSR was because they did not understand the labelling scheme:
“I don’t really understand the star ratings so that would be very low on my list”“I don’t know if only some brands do it and others don’t”“You’re right it’s really inconsistent isn’t it”“You don’t necessarily know what it means”(group 1 participants)

Instead, they deferred to the more familiar traffic light system used in school canteens.
“I’d love Australia to have the traffic light system of rating. I think some countries have it, it’s mandatory to have it on all the boxes and it basically rates the food as to how healthy it is”(group 1 participant)

### 3.3. Feeding Them Well Versus Keeping Them Happy

Participants allowed their children to eat some unhealthy foods or meals, provided they ate more healthy foods or meals on balance, as discussed by some of the group 2 participants:
“I know they’re eating wholegrain bread for lunch and they’re going to be having chicken and veggies (sic) for dinner, so I can make a decision to give them something high in sugar”“The 80:20 rule”“Plus the kids will probably burn it off as well, if you give them a little bit now and then it’s not too bad”(group 2 participants)

Some participants allowed their children to eat ‘junk’ occasionally as they ate healthily for the rest of the week. They described the balance between ensuring their children ate well, and making compromises to meet the challenges of everyday life.
“Sometimes I’m busy and I might have nuggets in the freezer so we’ll chuck them in the deep fryer”“The whole point for me is it’s too difficult to say no to my kids all the time”(group 5 participants)

However, participants in group 5 also expressed a need to exert control over foods their children ate. They used quite forceful language to express the importance of making responsible decisions, openly describing how their children regularly influenced their food purchase decisions.
“I mean I dictate a fair bit about what I buy, and I have been made to buy Dora [the explorer, cartoon character] baked beans over normal ones”“I’m a little bit strict”“My kids don’t dictate to me at all, what I say goes, but it’s not like their plate has to be cleared, if I think they’ve had enough and they’ve eaten a substantial meal they can go”“I’m cool to say no, but the boys like to have crisps, and probably twice a week I let them, that’s a treat for them because they’ve been good”(group 5 participants)

Participants in group 3 described how they enforced some rules about appropriate foods for their children by restricting the foods they bought.
“I don’t have any [junk] in my house”“We don’t buy any [junk]”“No it doesn’t come into my house”“There’s no soft drinks in my house”(group 4 participants)

### 3.4. Lack of Certainty in Packaging Information

Participants’ responses indicated they lacked certainty in information about diet and health coming from many sources, as highlighted by a discussion from group 2:
“You can burn off fat better than you can burn sugar”“I’m way more concerned about sugar than fat these days”“The trans fats are supposed to be shocking, and they have to put more sugar in to make the trans fats taste good”“I’ve no idea what that is”(group 2 participants)

The fundamental question of what constituted a healthy diet was discussed by some participants who were unsure what to look for on packaging. When group 5 participants were tasked with sorting the packaging used as visual stimuli, there was a lengthy discussion about how they would identify healthy products, which one participant summarised:
“Are we defining healthy as sugar, fat, salt? Or are we defining healthy as chemicals, additives, preservatives?”(group 5 participant)

They acknowledged that information was available to assist them, provided they knew how to read it. They stated a lot of people do not know how to read the nutrition information present on food packaging, which required consumer education to understand. Even if able to understand packaging information, the time needed to make an informed food choice was not practical for most participants, who simplified the process by looking for specific information as described in the ‘Desire to speed up shopping’ theme. However, a few participants accepted packaging information and claims without question.
“If they say it’s good for me I’ll think it is, I don’t like to put too much thought into it because it really confuses me”(group 2 participant)

HSR was discussed as an option that could facilitate removal of unnecessary information on packaging. Many participants agreed that one simple indicator would assist them to select healthy foods, provided the other information was removed.
“I think the stars is better than all that writing, a quick summary”(group 1 participant)

However, group 4 had a discussion that showed their confusion, and lack of trust in the HSR.
“The more stars the more nutritious the product, now what does nutritious mean?”“I fail to see how that [product] can be healthy when it’s full of sugar”“It just says more nutritious, but what does it actually take to get a 5 star rating?”“What does more nutritious mean?”“It’s a sort of grading, it depends on how healthy [it is]”“It doesn’t actually tell you what you have to do to get that rating, does it anywhere on [the packaging]?”(group 4 participants)

Participants said that supermarkets would resist assisting them to select healthy foods, arguing that information was already clearly displayed. However, navigating information in the supermarket environment is something participants do want help with, as expressed by one participant:
“It is very, very tricky, very tricky, that’s why you’ve just got to keep it simple”(group 5 participant)

### 3.5. Government is Trusted and Should Take Charge

When discussing personal responsibility for making healthy food choices, participants stated it was difficult to make good decisions without information from a trusted source, such as the government. There were no complaints of the ‘nanny-state’ telling parents what to do. Some protested that the government should do more to assist parents to select healthy foods, whilst others did not believe the government was empowered to do any more. Participants in group 3 discussed how parents were ultimately responsible for what their children ate, but they needed help:
“It’s your decision you know at the end of the day, but I do think there are ways and [government] regulations that could help people”“I think the onus is on us, but I think it would be helpful if it was easier to interpret the information”“The government should be doing more about the labelling so that it’s clear and concise instead of putting pressure on the parents to put healthy stuff in the lunchbox. It should be the parent’s responsibility to eat healthily at home, it should be the government’s responsibility to protect people’s inundation with misleading information”(group 3 participants)

Participants from all focus groups discussed how the government should assist them to make healthy food choices by telling manufacturers what information to display on packaging. They described how the government’s role was to establish labelling requirements that show the healthiness of foods and monitor food companies’ compliance with the guidelines, instead of relying on voluntary policies.
“I think if [food manufacturers] didn’t have to they wouldn’t put [any information] on there… if they didn’t have to they wouldn’t tell you how much sugar was in there”“It’s totally the government’s responsibility”“The government doesn’t care about our health”“Okay so the government needs to make them [food manufacturers] do it, make them do it that’s right, then check they’re doing it right”(group 2 participants)

Many participants from across all focus groups believed labels would only be accurate and helpful with government leadership. They were concerned that food manufacturers would be selective with labelling information unless government mandated what should be included.
“They’ve got the responsibility for making sure that the nutritional information is correct, and prosecuting and punishing offenders who’ve lied about their nutritional information”“I think they should make it clearer, I still don’t know how much is too much… maybe asking the manufacturer to make it clearer”(group 4 participants)

The level of implicit trust in the government was remarkable. For example, there was some awareness of HSR but trust in the scheme increased as they discovered the government’s role. For example, participants in group 1 read some of the HSR information written on the packaging stimuli:
“It’s the content of the food at a glance: to help make informed food purchases and healthier choices the more stars the more nutritious the product”“So it’s by the government”“It’s a summary they’ve just condensed into one thing”“They should say to the mothers out there that this is the system that we are now using like the Australian-made logo and just letting us know what it actually means, it’s a great idea”“I think it’s better than all that writing, a quick summary”(group 1 participants)

The question of whether the voluntary HSR system would assist them to make healthier food choices was discussed by several groups. The view that the government needed to make HSR mandatory to be effective was expressed repeatedly, as summarised by a group 4 participant:
“In a perfect world I think actually if somebody could decide on a methodology whether it’s ticks, whether it’s stars then it would be government’s responsibility to implement that kind of thing, I don’t think we should have voluntary things which you know the only people who volunteer to be part of it are the ones who know they’re going to get a good rating”(group 4 participant)

### 3.6. Food Manufacturers’ Health Messages Are Not Trusted

Participants described how they tended to choose products made by specific brands; however, they stated this was due to preference for the products and not out of any brand loyalty. They described a general lack of trust in food manufacturers to assist them to select healthy foods, due to the profit-driven nature of food manufacturing, as illustrated by a discussion amongst group 5 participants:
“I don’t trust a lot of the labelling, I feel like a lot of the labelling is marketing”“They’ll put [snack food] in the green wrapper and then they’ll put a picture of a banana on there and then they’ll say it’s fat free, but it’s full of sugar”“See that’s the thing you look at something like this [product] and it’s got no artificial colours, additives, but it’s full of salt”“You’ve got to know what it’s saying”“I think the manufacturer has some responsibility as well”“There’s guidelines in place, and I think [food manufacturers have] got a responsibility to communicate clearly to us what’s there”(group 5 participants)

Participants expressed frustration at the lack of transparency on food labels and had difficulty understanding health-related packaging information. Some participants described how they believed that food manufacturers omitted important information and gave examples of ways packaging information was misleading or deceptive. Groups 2 and 5 discussed use of per cent daily intake values for adults on packaging targeting children as a particularly concerning example of misleading labelling.
“My concern is that when they say this is 5% of your daily intake, it’s based on an adult male average diet, and these foods are aimed at children, and have different nutritional requirements, so it’s not 5% of their fat intake for the day, it’s who knows what percentage it should be”“I think the companies should be honest about what they put in it”“Do the companies care about our health? Or do the government care about our health?”“Like you say with this here it says 90% of your daily intake, but this is aimed for a kid and then it’s based on the average adult diet”“It’s misleading definitely”“I think every product here has some sort of misleading information on it”“But they get around it somehow”“Yeah, it’s not untrue it’s just misleading”(group 2 participants)

Some participants also challenged serving sizes which were not considered to reflect the amount people usually eat and the use of selected nutrient claims that failed to reflect overall nutritional quality.
“100 grams is less than a cup and you’d give your child more than a cup worth of cereal”(group 3 participant)

There was acceptance that food manufacturers are commercial enterprises, and that packaging was mainly used for marketing purposes. Participants believed manufacturers would not act responsibly of their own accord and did not trust them to determine what constitutes healthy food. Every group discussed how food manufacturers should follow government guidelines, and provide transparent information to consumers.

### 3.7. Supermarkets Slkhould Assist Parents to Select Healthy Foods

Most participants, who shopped in supermarkets rather than online, described shopping with their children as stressful due to practical considerations of caring for and managing young children, as well as managing responses to children’s demands for foods they found appealing, as illustrated by one participant’s comment:
“All three of the kids have a little basket and they continually run into old people, and I fill them up, and then they argue over who gets to carry what, and I try not to shout at everyone”(group 5 participant)

Participants used a number of strategies to select healthy food as described in the previous themes, which included selecting products they were already familiar with. Whilst supermarkets were not held responsible for the nutritional quality of foods sold, most participants viewed them as service providers and expected high standards. Most participants discussed how supermarkets should make it easier to find healthy foods, as summed up by one:
“They are so big now that they should be able to, if they want us to come and trust their shops then why can’t they?”(group 1 participant)

Participants from all of the groups suggested ways supermarkets could assist them to select healthy foods, and examples included having specific locations for displaying healthy choices:
“[Products] should be categorised a bit more than what they do now, as to what’s healthy and what’s not”“I think [they should put] specific healthy foods in specific sections so that you know that you’re getting less sugar and less artificial additives, and then having the rest next to them but with some sort of division, so that could help people that have less time or don’t want to look at the nutritional side of things and be more healthy, even use the school canteen [traffic light] system, it’d help wouldn’t it?”“I reckon have all the healthy stuff at eye level and then all the junk food and stuff down where you don’t look”(group 2 participants)

Introducing shelf-labelling to identify healthy products was discussed by group 4 participants:
“The supermarket could so easily just hang a little tag next to it, a little green light, and explain what falls into those categories so you can go and make a decision about it, well it’s just an easier shopping decision isn’t it?”“That’s relying on you trusting that system”“That sort of stuff should be in the ‘more info’ button… and you click to see more info and get the details online”(group 4 participants)

Some participants said supermarkets had taken advantage of consumer demand for foods they considered to be healthy by charging more. Most participants did not believe supermarkets would introduce measures to assist them for commercial reasons. They anticipated supermarkets would argue that provision of product information, alongside selling healthy foods such as fresh produce, was sufficient to assist consumers to select healthy foods, as illustrated by two group 3 participants:
“Woolies and Coles [supermarket chains] are there to have everything in the one place for you and I’d love it if they made things really easy and they helped but in a consumer money driven world it’s unlikely”“Woolies is unlikely to go through the 3 million products they have in store and guide you through, they can say that the information is clearly displayed on pretty much every item they have”(group 3 participants)

There were some additional frustrations from the online shoppers from group 3 about supermarkets’ websites. They described how absence of ingredients lists and nutrition information made it impossible for participants to make healthy choices.
“I get very frustrated because I will see something on special and you can’t go in to see the ingredients, oh well even if it’s a really good price I won’t put it in my shopping cart”“I shop online and a lot of things don’t have ingredients on there, so I just won’t buy them unless it’s something I’ve had before and I know it’s OK”“Yes, [those details] should be online, because I get frustrated with it all the time”“They could make it a bit easier to navigate too. You can’t search for no preservatives, or no added sugar, you can’t even search for nut free….”“Online sounds like way more trouble than its worth”(group 3 participants)

When discussing who should assist them to select healthy foods when shopping, participants stated that supermarkets had a responsibility to provide good service to their customers which included assisting them to select healthy foods.
“If you are online and you can’t find the information you want, I think they are doing themselves a disservice because you are not going to shop there”“They are providing a service, so I think anyone who is providing a service does have a responsibility to their customers to provide the very best service that they can”(group 3 participants)

## 4. Discussion

This exploratory study describes how packaging information, a key marketing method, impacts parents’ ability to select healthy foods in the supermarket. The challenges faced when selecting healthy foods were described, and the concept of voluntary policies was explored as parents described how government, food manufacturers and supermarkets should assist them. A political and ethical lens was applied to analysis, whereby large companies accept responsibility for their impact on society as an ethical obligation. The study shows how the voluntary information provided on food packaging, which is the result of the dominant neoliberal political agenda, restricts consumers’ ability to select healthy foods.

### 4.1. Parents’ Ability to Navigate Marketing Information on Food Packaging to Select Healthy Foods

Participants described shopping in supermarkets with young children as stressful and limiting their ability to select healthy foods. These findings are consistent with American research, which found parents rarely had enough time to evaluate products in order to make healthy choices [12], and a New Zealand study, which found labelling information had little influence on parents’ food choice due to competing demands [58]. Multiple demands required participants to select food their families would eat and schools would permit in lunch boxes, and provide a healthy diet, on a budget. They described the balancing act between allowing their children to exert influence over food selection and the need to select appropriate healthy foods. In short, making healthy food choices was just one of a multiplicity of criteria they considered.

Most parents in this study displayed an authoritative feeding style, whereby they encouraged healthy eating, whilst also allowing their children some influence over the foods they ate. Comparison of the three main parental feeding styles (i.e., authoritarian, permissive and authoritative [59]) has shown that children of parents with an authoritative style are more likely to have healthier eating habits [60] and a healthy weight [61]. Even though quite forceful language was used by some parents to describe how they exert control over the foods their children ate, compromise was also evident. The most effective strategy the parents appeared to use was to avoid having the foods they considered inappropriate, such as soft drinks, in their homes.

The lack of certainty in packaging information experienced by participants also limited their ability to select healthy foods. It was evident that routine decisions about food choice were not necessarily made using the information presented on packaging. It is likely that this is due to ‘information overload’, which occurs when the amount of information available becomes more of a hindrance than a help [62]. Food labels can help consumers to make informed dietary choices; however, even when the information is accurate and reliable, it may not be widely understood [63]. Previous Australian research has reported the difficulty parents experience when interpreting food labels to identify healthy choices [64]. The time required to read and understand food labels, particularly for people with low levels of food literacy, can deter them from using this information [65]. Feelings of time scarcity can be mediated by food literacy or nutrition knowledge, but many people will automatically substitute a shorter activity (e.g., using colourful packaging as a heuristic cue) for longer ones [66]. This is despite the importance placed on health and nutrition when selecting food for their children [67]. Consumers respond to information overload on packaged products by using a limited number of attributes [68], which was evident as participants described making judgements using their own criteria, or structural cues such as avoiding the confectionery aisle. It is for this reason that the government-led HSR was developed.

The Australian government-led HSR nutrition label aims to assist consumers by providing a single, consistent, accurate, front-of-pack labelling system [41]. Products are scored from ½ to 5 health stars (the healthiest choice), based on an algorithm that allocates points for positive components and subtracts them for negative nutrients [69]. Participants had low awareness and understanding of HSR, which was not surprising as only a thousand packaged foods carried the rating at the time of the research, 15 months after implementation [70]. Findings suggest that for consumer trust in HSR to be achieved there needs to be greater transparency regarding who is responsible and the nutrition criteria applied, consistency with more established schemes (e.g., school canteen traffic lights) and familiarity, which can be built over time [71]. The HSR, and similar schemes, would have greater impact as part of a broader range of policies designed to create healthy food environments [71].

### 4.2. Who Parents Thought Should Be Responsible for Helping Them to Select Healthy Foods

#### 4.2.1. Government’s Role

The government was trusted by this study’s participants, who thought they should take more responsibility. They expected the government to set the rules for health statements and claims that food manufacturers could then use on packaging, and monitor compliance. Participants believed the government should set the criteria used to define healthy foods, not food companies. Trust in HSR increased when participants discovered the scheme was government-led. These findings suggest parents would support a regulatory approach to provision of nutrition signposting on packaging, rather than information-based voluntary policies. This is consistent with a government survey of Western Australian adults, which found that 97 per cent believed regulating nutrition information on food labels was important [72]. South Australian consumers also held the government responsible for food labels; however, in contrast to this study’s findings, they distrusted the government’s ability to act responsibly [73]. The Australia New Zealand Food Code sets criteria that are required to be met for health and nutrition claims [74]; however, marketing statements and claims, which are more prevalent, are not subject to the same level of scrutiny [22].

#### 4.2.2. Food Manufacturers’ Role

Food manufacturers’ voluntary efforts to assist consumers to select healthy food were not trusted by participants, nor did they trust them to determine what constituted healthy foods. In fact, they said oversight by the government was needed. From an ethical perspective, participants believed voluntary policies by food manufacturers were influenced by commercial interests, which is consistent with South Australian research [73]. Similarly, European consumers were mistrustful of nutrition claims on food products and required more information before making a choice [75].

Participants assumed food manufacturers would be unwilling to voluntarily highlight unhealthy foods. For example, they would only apply HSR to products that achieved a good rating. This is concerning, as it supports the ‘health halo’ theory, which asserts that the presence of a health message implies the food is healthier than it actually is [76]. This study’s findings indicate voluntary policies from food manufacturers that include providing specific packaging information were not effective in assisting parents to select healthy foods, mainly because the companies were not trusted.

#### 4.2.3. Supermarkets’ Role

Supermarkets were more trusted than food manufacturers by this study’s participants. Despite describing a definite role for government in setting rules for food manufacturers, the need for the government to monitor supermarket activities was not discussed. Supermarkets’ failings were expressed as frustrations rather than examples of deception, including failure to display product information for online shoppers, which is a gap in current food regulations [77,78]. Supermarkets have been described as trusted food authorities due to long associations with experts such as the Dietitians Association of Australia, as well as celebrity chefs such as Jamie Oliver and Curtin Stone [79]. The public health impact of supermarkets has been overlooked by governments [80], who seek their presence during policy making. This is concerning because supermarkets hold a powerful position in the Australian food system, and few positive public health impacts have been identified to date [3].

Participants proposed ways supermarkets could assist parents to select healthy foods. Suggestions included introducing shelf-labelling to identify healthy products, and supermarkets’ dedicated health food aisles were referred to as a location for healthy choices. These suggestions indicate supermarkets need to take a broader structural approach to assist parents to select healthy food, not just food labelling. Australian supermarkets’ so-called health food aisles currently sell processed packaged gluten-free, vegetarian and organic foods, as well as whole foods including nuts and grains, with no indication of criteria used to designate health status [81,82]. Redesigning the way products are displayed in supermarkets, and increasing the availability of healthy foods by allocating more shelf-space have been identified as important [83]. However, participants believed supermarkets would not take action for commercial reasons.

Previous research has concluded that Australian supermarkets are less active than food manufacturers and food service operators in specifying and implementing voluntary policies to assist customers to select healthy foods [31], which presents an opportunity for Australian supermarkets. A review of the voluntary policies that can impact public health made by 31 of the world’s largest supermarkets revealed some action has been taken [84]. Approaches include: restricting multi-buy promotions that encourage bulk purchase of unhealthy foods [85]; removing lunchbox-sized sugar sweetened drinks from sale [86]; introducing a supermarket-wide shelf-edge labelling system that identifies healthy foods [87]; introducing personalised shopper profiles that track the amount of healthy foods purchased [88]. Supermarkets’ ability to implement voluntary policies to assist parents to select healthy foods deserves further examination.

### 4.3. Strengths and Limitations

There are strengths and limitations to this study. The focus groups provided insight into participants’ use and understanding of packaging information, and who they held responsible for providing it. Focus group participants were purposively recruited to ensure the views of parents from higher and lower SES areas were represented, and assigned to focus groups based on their SES classification. Many of the participants were well-educated, from two-parent households, in medium to high SES neighbourhoods, which may have influenced results. Nevertheless, the main themes identified were consistent across all focus groups. The questions were asked to explore consumers’ indirect impressions of voluntary policies as there appears to be little direct awareness. Further research into the ability of voluntary policies to support consumers to select healthy foods is recommended. Findings cannot be generalised to the broader population, or for supermarket shopping where there is less time to consider packaging information. The high market-share products used as stimuli may not normally be purchased by the participants. The views of food system actors discussed by participants (i.e., food manufacturers, government, supermarkets) were not considered, and future research examining their perspectives would add to the literature.

### 4.4. Implications for Public Health Policy and Practice

The findings from this study indicate that the current Australian government response to food marketing to children, which favours information-based voluntary policies, does not assist parents to select healthy foods. The government-led voluntary HSR is one of only two national policy actions, the other being the Healthy Food Partnership [89], a public-private-partnership which aims to improve the nutritional health of all Australians [26]. Former Australian food policy interventions have failed to deliver integrated food policy (the National Food Plan [90]), or proved to be inefficient and unsustainable (the Australian Food and Health Dialogue [91]). Providing information and placing responsibility on individuals is common government policy in other countries [92], but regulatory approaches are more effective [27].

A number of government-led policy actions are suggested as a result of this study, including: consider the public health impact of supermarkets’ business practices; develop initiatives that go beyond provision of nutrition information to include other aspects of food environments (i.e., food availability, price, promotion and placement); require online food retailers to display product information; restrict marketing techniques on packaging of poor nutritional quality foods; communicate existing guidance on healthy eating and food selection.

A structural approach for supermarket action is recommended, as they appear to be trusted and have the power to assist parents to select healthy foods [3]. Supermarkets that accept responsibility for their impact on society as an ethical obligation can assist parents to select healthy foods by: applying appropriate health criteria for foods sold in health food aisles; placing healthy foods in prominent locations; introducing shelf-edge labels to identify healthy choices; providing product information for online shoppers.

## 5. Conclusions

The findings from this study indicate that voluntary policies relating to provision of labelling information, which is a typical policy response to food marketing targeting children in many countries, does not assist consumers to select healthy foods in real world settings. Parents consider a multiplicity of criteria when selecting food, and struggle to navigate food packaging marketing techniques. The government should take action to build trust in the HSR, restrict marketing techniques present on the packaging of poor nutritional quality foods and communicate existing dietary guidance on healthy eating and food selection. Food manufacturers should demonstrate ethical consideration of their impact on society by providing transparent packaging information. Supermarkets should take a structural approach to voluntary policies to assist parents to select healthy foods. Current supermarket action deserves scrutiny, as they appear to have established some trust with consumers.

## Figures and Tables

**Figure 1 ijerph-16-03377-f001:**
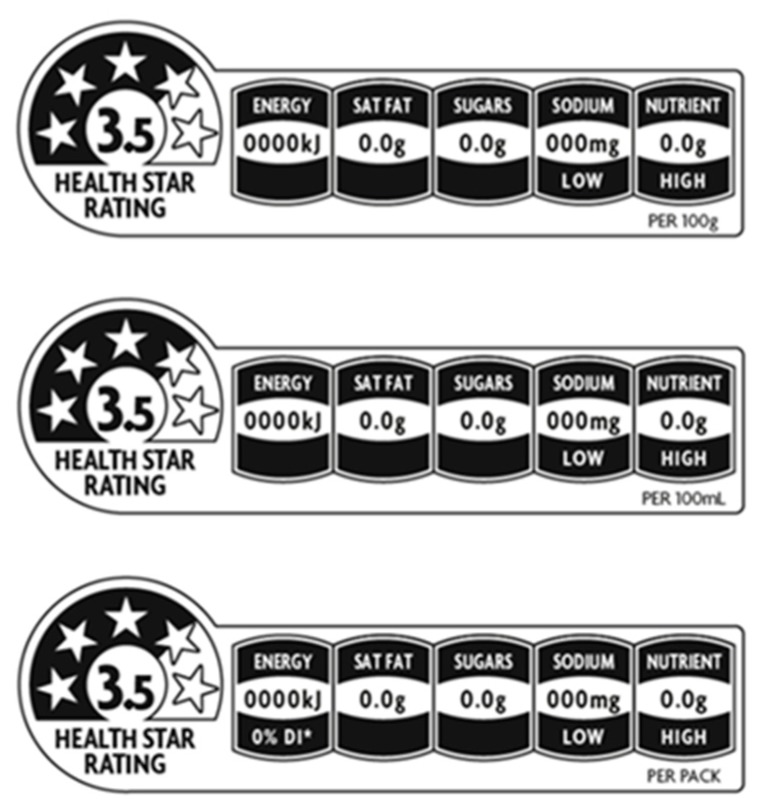
Health Star Rating nutrition label [30].

**Table 1 ijerph-16-03377-t001:** Corporate social responsibility commitments of selected high-market-share Australian food companies.

Food Company ^1^	Importance of Nutrition and Health
Nestle [42]	Nestle’s 10 principles of business operations places nutrition health and wellness first. “Our core aim is to enhance the quality of consumers’ lives every day, everywhere by offering tastier and healthier food and beverage choices and encouraging a healthy lifestyle. We express this via our corporate proposition Good Food, Good Life.”
Kellogg’s [43]	Kellogg’s seeks to nourish families so that they can flourish and thrive. “We believe nutrition literacy is crucial in helping consumers make informed food choices for themselves and their families. Through on-pack labeling and website content, we provide comprehensive nutrition and ingredient information, including details on calories, fiber, fats, sugar and other nutrients, for all of our foods.”
Sanitarium [44]	Sanitarium’s promise to consumers: “One of Sanitarium’s core philosophies is truly nourishing food, and each Sanitarium product is designed to meet our high nutritional and food appeal standards. We invest significantly in providing the community with free nutritional information and advice through our team of qualified nutritionists.”
Woolworths [45]	Woolworths’ corporate information does not include commitments on health and nutrition. However, when announcing their partnership with Jamie Oliver [46], they stated: “The partnership will focus on bringing better, healthier, affordable fresh food to life for everyday Australians, giving them the information and confidence to prepare great tasting fresh meals at home.”

^1^ The cited reports were current at the time of conducting the focus group discussions.

**Table 2 ijerph-16-03377-t002:** Focus group sample characteristics.

Sample Characteristics	Total
	*N* = 37
Gender	
Male	4
Female	33
Age group	
18–25 years	2
26–35 years	13
36–50 years	22
Socioeconomic status	
Low	14
Medium to high	23
Age of youngest child	
Preschool (2–4 years)	23
School age (5–8 years)	14
Gender of youngest child	
Male	21
Female	16
Number of children	
1	9
2	21
3	6
>3	1
Highest level of education	
Year 12	5
Trade/diploma or TAFE course	15
University Bachelor degree of higher	17
Marital status	
Married	27
Defacto	8
Divorced/ separated/widowed	2
Employment status	
Self-employed	5
Employed part-time	13
Employed full-time	5
Home duties	11
Student	1

Footnote: TAFE is Technical and Further Education.

**Table 3 ijerph-16-03377-t003:** Questions used to promote focus group discussions.

Questions	Notes to Guide the Discussion	Research Question Addressed
1. Introductions and icebreaker.	Lead the introductions, providing an opportunity for the participants to meet each other and feel more comfortable about joining in the group discussion.	
2. Looking at some examples of foods from the supermarket, do you recognise any of them?		RQ1
3. Thinking about shopping in your regular supermarket, what information do you use to decide if foods meet you or your children’s needs?	If health isn’t brought up by participants, return to this question after asking Q4 and ask: Where do you get the information you need to decide if the foods are healthy?	RQ1
4. Who is responsible for providing that information?	Do food companies or supermarkets have any responsibility? If health isn’t brought up by participants, return to this question and ask: Who is responsible for providing you with the information you need when shopping, to decide if foods are healthy? How do you think they are doing?	RQ2
5. Is there anything you really like about the information food companies put on food packaging?	Is there anything you really don’t like?	RQ1
6. Returning to the examples of foods from the supermarket, can you sort these foods into groups of similar products?	Ask the group to sort the products without giving them any criteria, to explore how they categorise the foods. If health isn’t used as the main sorting characteristic, ask the group to sort the foods again, this time based on perceived healthiness, and talk about the reasons why they put them into the selected category.	RQ1, RQ2
7. In your opinion, is there anything that food companies or supermarkets should be doing to help you choose healthy foods for your children?		RQ3

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
