# Peer review of "The Impact of Voluntary Policies on Parents’ Ability to Select Healthy Foods in Supermarkets: A Qualitative Study of Australian Parental Views"

_ijerph, 2019, doi:10.3390/ijerph16183377_

Round 1

Reviewer 1 Report

The manuscript explores how the health star labels influences purchase, the manuscript is well written but lacks structure especially on some points in the results and discussion section which needs some minor revision.

Figure 1. These are triplicate figures, is there a reason why the same three are shown here?

Table 3. Consider adding a header here, it's a bit sparse on what the general theme idea that is being asked. Or a framework perhaps can be included here.

Discussion seems very sparse and needing more work, perhaps consider sub sectioning it to based on the results heading on the previous section. Currently it's difficult to read as the authors are jumping from one idea to another.

Reviewer 2 Report

This qualitative study explored the Australian parents’ views of the impact of voluntary Health Star Rating policy on their ability to purchase nutritious foods in supermarkets. Before it is considered for publication, I would ask the authors to make the suggested changes in the manuscript.

Line 20 – Specify the number of fathers and mothers

Line 21 - Group discussions were audio-recorded…….

Line 22 – Specify whether a software was used to carry out the analysis or manual analysis

Line 43 – Name the nutrient profiling tool

Line 45 – Replace food with foods

Lines 54-56 – You may link overconsumption of nutrient-poor foods to increasing prevalence of obesity in children

Line 64 – Besides HSR, provide examples of other voluntary policies - I did see the mention of one in Discussion section but I would suggest to discuss it in Introduction section

Line 90 – Briefly explain HSR

Line 109 – Did you use any paradigm (e.g. social constructivism) to inform your qualitative study? Discuss

Line 141 – I am a bit curious about the duration of 5 FGDs. All the FGDs were completed in 90 minutes. There were no variations in the duration???

Line 179 – Table 3 - What was the source of questions included in the interview guide? Were the questions pre-tested?

Line 202 - Few examples of coded text segments could be given for better understanding of the thematic process

Line 348 – Instead of ‘lack of certainty in packaging information’ you may consider the theme of ‘lack of food literacy’

Lines 623-624 – I am not sure about the use of following sentence here “In Britain…… brand foods.” It does not fit well here.

In Discussion, authors must discuss about the parental control over consumption foods in children i.e. use of forceful language - discuss about authoritative style etc. This will strengthen your discussion section. Similarly, the authors mentioned about the difficulty to navigate food packaging in results section, but they need to discuss this in the context of lack of food literacy.
